# Wastewater Speaks: Evaluating SARS-CoV-2 Surveillance, Sampling Methods, and Seasonal Infection Trends on a University Campus

**DOI:** 10.3390/microorganisms13040924

**Published:** 2025-04-17

**Authors:** Shilpi Bhatia, Tinyiko Nicole Maswanganye, Olusola Jeje, Danielle Winston, Mehdi Lamssali, Dongyang Deng, Ivory Blakley, Anthony A. Fodor, Liesl Jeffers-Francis

**Affiliations:** 1Biology Department, College of Science and Technology, North Carolina A&T State University, 1601 E. Market Street, Greensboro, NC 27411, USA; sbhatia@aggies.ncat.edu (S.B.); ojeje@aggies.ncat.edu (O.J.); dmwinston@aggies.ncat.edu (D.W.); 2Built Environment Department, College of Science and Technology, North Carolina A&T State University, 1601 E. Market Street, Greensboro, NC 27411, USA; mlamssali@aggies.ncat.edu (M.L.); ddeng@ncat.edu (D.D.); 3College of Computing and Informatics, University of North Carolina at Charlotte, 9201 University City Blvd, Charlotte, NC 28223, USAafodor@charlotte.edu (A.A.F.)

**Keywords:** wastewater-based epidemiology, SARS-CoV-2 RNA detection, university public health surveillance, grab and composite wastewater sampling, PMMoV normalization

## Abstract

Wastewater surveillance has emerged as a cost-effective and equitable approach for tracking the spread of SARS-CoV-2. In this study, we monitored the prevalence of SARS-CoV-2 on a university campus over three years (2021–2023) using wastewater-based epidemiology (WBE). Wastewater samples were collected from 11 manholes on campus, each draining wastewater from a corresponding dormitory building, and viral RNA concentrations were measured using reverse transcription-quantitative PCR (RT-qPCR). Weekly clinical case data were also obtained from the university health center. A strong positive and significant correlation was observed between Grab and Composite sampling methods, supporting their robustness as equally effective approaches for sample collection. Specifically, a strong correlation was observed between Aggie Village 4 Grab and Aggie Village 4 Composite samples (R^2^ = 0.84, *p* = 0.00) and between Barbee Grab and Barbee Composite samples (R^2^ = 0.80, *p* = 0.00). Additionally, higher viral RNA copies of SARS-CoV-2 (N1 gene) were detected during the Spring semester compared to the Fall and Summer semesters. Notably, elevations in raw N1 concentrations were observed shortly after the return of college students to campus, suggesting that these increases were predominantly associated with students returning at the beginning of the Fall and Spring semesters (January and August). To account for variations in fecal loading, SARS-CoV-2 RNA concentrations were normalized using Pepper Mild Mottle Virus (PMMoV), a widely used viral fecal biomarker. However, normalization using PMMoV did not improve correlations between SARS-CoV-2 RNA levels and clinical case data. Despite these findings, our study did not establish WBE as a consistently reliable complement to clinical testing in a university campus setting, contrary to many retrospective studies. One key limitation was that numerous off-campus students did not contribute to the campus wastewater system corresponding to the monitored dormitories. However, some off-campus students were still subjected to clinical testing at the university health center under mandated protocols. Moreover, the university health center discontinued reporting cases per dormitory after 2021, making direct comparisons more challenging. Nevertheless, this study highlights the continued value of WBE as a surveillance tool for monitoring infectious diseases and provides critical insights into its application in campus environments.

## 1. Introduction

SARS-CoV-2, a zoonotic virus that was initially identified in China in December 2019 and spread globally at an alarming rate, is now considered the deadliest virus of the 21st century [1,2]. COVID-19 is spread via direct exposure through respiratory droplets with individuals in proximity and indirect exposure through surface contact with fomites [3]. Although most COVID-19 exposures have been reported to occur via the respiratory system; recent research has revealed that SARS-CoV-2 RNA is found in the gastrointestinal tract [4,5]. Recent studies worldwide have indicated the detection and quantification of SARS-CoV-2 through sewage and wastewater [6,7,8,9,10]. Viable SARS-CoV-2 is released through human feces, saliva, and sputum and makes its way through wastewater [4,11].

Wastewater is a composite biological mixture of an entire community with biological specimens of each person in the community accumulated each day [12]. Wastewater-based epidemiology (WBE) is traditionally used for testing illicit drugs and sexually transmitted diseases (STIs) within communities [13,14] and has, more recently, gained traction in detecting SARS-CoV-2 RNA in wastewater. It has been utilized for detecting enteric viruses such as poliovirus, norovirus, and hepatitis A virus [15,16]. Detection of SARS-CoV-2 in wastewater has become a novel approach to monitor viral shedding dynamics and make informed public health decisions [17]. SARS-CoV-2 has been shown to shed into feces in 27–89% of infected individuals, with estimated genomic copy levels ranging from 10^2^ to 10^7^ gc/mL in feces and from 10^2^ to 10^5^ gc/mL in urine [18,19]. The average duration of SARS-CoV-2 RNA shed in feces is approximately 17 days; however, this duration may vary as new viral variants emerge [20].

Wastewater-based epidemiology (WBE) studies have reported evidence of local community transmission of SARS-CoV-2 prior to the first clinically documented cases [8,9,21,22]. Several studies in the United States found that WBE data predicted new clinical case reports by 2–8 days [17,23] and another study reported that viral titer trends in wastewater preceded clinical data by 4–10 days [24]. These findings suggest that monitoring SARS-CoV-2 RNA in wastewater can serve as an early indicator of the virus’s presence in a community. Furthermore, a real-time study conducted in California found a correlation between SARS-CoV-2 RNA levels in settled solids and reported COVID-19 clinical cases [12,25,26]. Therefore, WBE has proven to be a valuable epidemiological tool and suitable complement to clinical testing, particularly in identifying delays in clinical testing within a particular community and underreported cases (along with increases in false positive and false negative results), facilitating early detection and outbreak prevention [27,28,29,30].

Real-time early warning, however, requires frequent wastewater sampling, rapid analytical methods, and timely communication of results to public health authorities. The correlation between SARS-CoV-2 RNA in wastewater and COVID-19 cases indicates that wastewater-based epidemiology (WBE) can offer valuable insights into community-level transmission dynamics. Moreover, WBE is not restricted to sampling at wastewater treatment plants (WWTPs). Consequently, larger communities, such as universities and hospitals, have implemented WBE to address gaps in clinical testing and enhance understanding of infection dynamics. University settings particularly offer a valuable opportunity to implement wastewater surveillance due to the significant presence of asymptomatic populations and concentrated areas of potential transmission. Numerous higher-education institutions and municipalities in the United States have recognized the utility of WBE as a surveillance tool for monitoring SARS-CoV-2 infection trends and swiftly identifying potential cases, thereby aiding health administrators in making informed decisions regarding infection control and contact tracing [19,31,32,33,34,35]. However, while most studies have focused on collecting wastewater from large wastewater treatment plants (WWTPs) or local sewer sheds, building-specific collection, particularly in campus settings, can also provide valuable insights for inferring clinical incidences in university populations.

Here, we reported the North Carolina A&T State University campus-wide wastewater surveillance analysis to monitor and assess SARS-CoV-2 transmission on campus. This study involved comparing SARS-CoV-2 concentrations in Grab samples collected biweekly from 11 different dormitories over a period of three consecutive years (Spring 2021–2023), alongside two Composite samples. To assess the efficacy of wastewater-based epidemiology (WBE) as a predictive tool for COVID-19 outbreak dynamics, we examined the correlation between the concentration of the SARS-CoV-2 N1 gene in wastewater and the clinical COVID-19 cases reported in the corresponding dormitories and campuses. Furthermore, we implemented a mixed linear effects model to evaluate the association between levels of SARS-CoV-2 in wastewater and clinical cases reported per dormitory on campus in 2021. Additionally, from Spring 2022 to 2023, we quantified wastewater fecal strength using Pepper mild mottle virus (PMMoV) as a normalization biomarker in every sample.

## 2. Materials and Methods

### 2.1. Sampling Sites

The study was conducted in 11 designated dormitories on NCAT campus (Aggie Village 1, Aggie Village 3, Aggie Village 4, Aggie Village 6, Aggie Village E, Barbee Hall, Cooper Hall, Haley Hall, Holland Hall, Morrison Hall, Morrow Hall, and Pride Hall) shown in Figure 1. The inclusion criterion for these dormitories was determined by their high population size, the placement of manholes in proximity outside the dormitory, resulting in a significantly higher inflow rate of wastewater into the manholes, rendering them particularly suitable for wastewater collection. Manholes placed outside the dormitories were chosen as the Grab sampling sites with untreated wastewater (Figure 1). For Composite sampling, two ISCO GLS autosamplers with a 7500 mL capacity were installed outside Aggie Village 4 and Barbee Halls, respectively, collecting samples every 24 h. Appendix A Table A1 provides details on each sampling site, including sampling frequency and sampling type.

### 2.2. Clinical Testing

NCA&T State University Health Center followed two clinical tests for COVID-19 diagnosis as recommended by CDC: antigen test via anterior nasal swab and Rt-PCR via nasopharyngeal swab samples. During 2020–2021, to decrease transmission, the institution required students who tested positive for SARS-CoV-2 to self-isolate in a quarantine dormitory (Haley Hall, ~average 10 students per month) and employed the wastewater-based surveillance (WBS) as a secondary screening tool for SARS-CoV-2 spread on campus since 2021. Positive COVID-19 case numbers for the State of North Carolina were retrieved from the NCDHHS dashboard [36]. The number of persons tested for SARS-CoV-2 on campus was compared to SARS-CoV-2 levels in the wastewater to estimate the number of RNA copies per person in total sewage water

### 2.3. Wastewater Collection and Pre-Treatment

Samples of wastewater were collected between January and December 2021, between January and December 2022, as well as between January and June 2023, utilizing wastewater streaming from campus buildings. Sampling frequency was twice a week in the morning and once a week during the Summer months. For sample collection, a new, sterile 2 L HDPE plastic bottle tied to a rope was lowered into the manhole sewer at each sampling site. Approximately 200 mL of grab wastewater and 24 h composite wastewater (from the ISCO GLS autosampler (ISCO, Inc., Lincoln, NE, USA)) was collected. All bottles were carefully transported to the laboratory at room temperature. Wastewater was aliquoted into 50 mL conical tubes and stored at −80 °C or processed for viral nucleic acid extraction.

### 2.4. Viral RNA Extraction

RNA was extracted according to the manufacturer’s procedure using Promega Total Nucleic Acid Wastewater Extraction kit [37]. Briefly, silica-based PureYield columns were used to absorb and concentrate total nucleic acids (TNAs) from large quantities of wastewater. Approximately 40 mL of protease-treated wastewater was used to extract total nucleic acid (TNA) using a 20 mL PureYield column. At the completion of the extraction, a total of 40 μL of TNA was eluted from each site (dormitory) in preheated (60 °C) nuclease-free water. The concentration and purity of the extracted RNA were determined by using a Nanodrop™ (Thermo Fisher Scientific, Waltham, MA, USA). The absorbance readings at 260 nm and 280 nm (260/A280 ratios) are commonly used to determine the purity of nucleic acid with a general acceptable range between 1.9 and 2.1.

### 2.5. Real Time-Quantitative Polymerase Chain Reaction

To detect and measure viral RNA, a SARS-CoV-2 RT-qPCR Kit for Wastewater was used (Promega; Madison, MI, USA). The primers for this kit were based on the published SARS-CoV-2 detection (Appendix A Table A2) and quantification from the US Centers for Disease Control [38]. For this study, the N1 gene (highly abundant) was targeted. The SARS-CoV-2 RT-qPCR reaction plate was set up with 6 μL of the RT-qPCR amplification mix and 4 μL of extracted nucleic acid, reference RNA, 0.5 μL of 20X primer/probe/IAC mix, and 0.3 μL nuclease-free water was used as a NTC (No Template Control)/negative control, with a final reaction volume of 10 μL. CFX Connect Real-Time PCR System (2021 samples) (Bio-Rad Laboratories, Inc., Hercules, CA, USA) and CFX 96 Touch Real-Time PCR System (Bio-Rad) (2022–2023) was used for RT-qPCR analysis following the thermal condition as 25 °C for 2 min for initiation, reverse transcription at 45 °C for 15 min, polymerase activation at 95 °C for 2 min, followed by running 40 cycles at 95 °C for 3 s and 62 °C for 30 s (CDC qRT-PCR panel 2020). Slopes and y-intercepts from RT-qPCR were used to quantify copies of SARS-CoV-2 in each reaction using the instrument’s recorded Cq value. Subsequently, the SARS-CoV-2 copy number value was then transformed into a gene copies/L of extracted wastewater value. Data was collected using the FAM/HEX/Cy5 channels of the CFX Maestro™ software 2.3 (Bio-Rad Laboratories, Inc.).

### 2.6. Quality Control

For quality control purposes, quantification of SARS-CoV-2 RNA was achieved using a standard curve with a dilution scheme (1:10) that was the same for the positive control and an NTC. For the standard curve, in vitro transcribed SARS-CoV-2 RNA (WA1-USA strain) fragments (RNA N + E) were used as a template (provided in the kit). It was generated by diluting the quantification standard RNA N + E (4 × 10^6^ copies/μL) 100-fold by mixing 2 μL in 198 μL nuclease-free water to obtain a final concentration of 1.6 × 10^5^ copies/μL. The Limit of Detection (LoD) was set at 8 copies per nucleic acid reaction and the Limit of Quantification (LoQ) was less than 25% coefficient of variance.

For 2022–2023 samples, the standard curve was generated by diluting the quantification standard SARS-CoV-2 RNA N + E (4 × 10^6^ copies/μL) 100-fold by mixing 2 μL, along with 2 μL of PMMoV virus RNA N + E (used as an internal control, 4 × 10^6^ copies/μL), in 196 μL nuclease-free water to obtain a total concentration of 1.6 × 10^5^ copies/μL.

### 2.7. Statistical Analysis

All statistical analyses were computed using Prism GraphPad v.9, SPSS v.25, or R v4.4.0. For the wastewater analysis, correlations were examined using wastewater N1 gene concentration, as well as SPSS v.25 and R studio. To determine if wastewater was a predictor of clinical cases, a cross-correlation between wastewater N1 gene concentrations per building and the average of all buildings with COVID-19 clinical cases on campus per semester was computed. Spearman’s (R_s_^2^) coefficient was determined for these analyses using base 10-logarithm-transformed values. These correlation coefficients were used to determine the strength of correlation relationship between the average wastewater N1 gene concentrations for a given date of specimen collection and the associated COVID-19 case rate, with a *p*-value of 0.05 or lower signifying significance. Correlations greater than 0.5 for R^2^ were considered strong. A one-way ANOVA was computed to quantify the differences between each dormitory and to test for statistical differences between sampling methods by comparing the detection of SARS-CoV-2 (Ct values) between samples collected using either the Grab or Composite method. All concentration of RNA copies/L determined for wastewater data between Spring 2021 and Spring 2023 graphs were generated using Prism GraphPad v.9 and R v4.4.0.

## 3. Results and Discussion

### 3.1. Clinical COVID-19 Positive Cases on NCAT Campus (2021–2023) and Correlation with State of NC

To determine the number of COVID-19-positive individuals in the dormitories, data was collected from the Student Health Center at North Carolina A&T State University (NCA&T) for the period spanning 2021–2023 (Spring, Summer, Fall), as shown in Figure 2a.

In the Spring 2021 semester, the health center recorded 175, 45, 166, and 314 COVID-19-positive cases in January, February, March, and April, respectively. During the Summer months of May to July 2021, only 23 positive cases were documented. However, a notable increase occurred in August, coinciding with the start of the Fall semester, with 312 positive cases reported. The Fall semester saw a gradual decrease in positive cases from September to December 2021. In 2022, a notable peak in cases was observed at the beginning of the year, with 290 cases reported in January, followed by 198 cases in April and 118 cases in August, contrasting with the low case numbers (49 cases) reported during the Summer months. Significant COVID-19 case numbers were recorded in January, April, and August, while the Summer months saw very low case numbers. The increase in cases in January and August can be partially attributed to the return of students from the Winter and Summer breaks, respectively, diminished vigilance regarding COVID-19 transmission risks, and the emergence of a more virulent Omicron subtype. These trends align with other studies observing increased SARS-CoV-2 prevalence in wastewater following holidays and gatherings, suggesting that post-travel and event-related surges in cases are common [39,40]. For Spring 2023, the highest number of cases occurred in February (246 cases), followed by January (126 cases), March (39 cases), and April (25 cases). According to the North Carolina Department of Health and Human Services (NCDHHS) wastewater dashboard, new case rates in Greensboro (North Buffalo WWTP) ranged from 3.75 to 1.62 per 10,000 persons between January and February, with a reported decline in March and April [41]. Overall, a decline in positive clinical cases was observed on campus from 2021 to 2023, likely due to prevalence of at-home testing and unwillingness of people to be tested often [42]. The data reflects clinical cases, with home test kit results not included in these figures.

### 3.2. Quantification of SARS-CoV-2 Viral RNA in Wastewater in Spring 2021–2023

To determine the presence of SARS-CoV-2 RNA in effluent wastewater from dormitories across the NCA&T campus during 2021–2023, we employed qRT-PCR for the quantification.

The results shown in Figure 3a reveal distinct variations in viral RNA levels during these periods. Higher viral RNA copies of SARS-CoV-2 for the N1 gene were observed during the Spring semester compared to both the Fall and Summer semesters of 2021. The average number of viral RNA copies detected in the Spring semester 2021 was 10^7^ log RNA copies/L, indicating a notable presence of the virus during this period. On average, the RNA copies in wastewater were higher during the Spring months (January–April) (1.81 × 10^5^) compared to the Summer months (May–July) (1.00 × 10^5^). This detailed seasonal comparison highlights the specific trends in RNA copies during Spring, Summer, and Fall. In 2022, higher viral RNA copies of SARS-CoV-2 for the N1 gene were detected during the Spring 2022 semester as compared to the Fall and Summer semesters in 2022. The average number of viral RNA copies detected in the Spring semester was 10^12^ log RNA copies/L with an average C_t_ value of 32.98 for three replicates (Figure 3a). The lowest average levels of 10^5^ log RNA copies/L were detected on campus during the Summer semester with an average C_t_ value of 34.64. The average C_t_ value for the Fall semester was 34.43 and average RNA copies detected were 10^11^, suggesting that the number of RNA copies detected was lower in the Fall semester than in the Spring semester of 2022. Differences in RNA concentration across dormitories are shown in Appendix A Figure A1. Higher viral RNA load was detected in Spring 2021 as compared to Spring 2022 and may be attributed to a higher number of vaccinated individuals as of 2022. Since the initiation of vaccination campaigns and the implementation of enhanced precautionary measures at the onset of the pandemic on campus, there has been a notable reduction in the number of clinical cases and RNA copies reported on campus since 2022 compared to the data from 2021.

Elevations in raw N1 concentration of SARS-CoV-2 were observed shortly after the return of college students to campus. We surmised that these increases are predominantly associated with the students’ return at the commencement of the Fall and Spring semesters (specifically in January and August). Comparable surges in SARS-CoV-2 RNA levels were noted at other university campuses throughout the United States, with data indicating that, upon the students’ return, SARS-CoV-2 RNA concentrations significantly increased in wastewater samples from the majority of student residence hall monitoring locations [32,33,34,43,44]. Additionally, the potential influence of seasonal variability on wastewater-based epidemiology (WBE) analyses remains a subject of investigation [45,46,47]. Seasonal temperature fluctuations primarily affect the in-sewer transit time of the target virus. A study by Hart and Halden [46] in Detroit indicated that the virus’s persistence was approximately 100 h in Winter and 20 h in Summer. Furthermore, the duration required for a 90% reduction in virus titer is prolonged at lower temperatures [45,47]. Consequently, the observed higher RNA copy levels during the Spring semesters of 2021 and 2022, compared to those in the Summer and Fall semesters, may be attributed to seasonal variability, warranting further investigation. The effectiveness of WBE is closely tied to the reliability and efficiency of analytical processes. Wastewater and sludge are characterized by complex sample matrices, and the concentrations of viruses or viral components are often extremely low, particularly during the initial phases of a community outbreak. Additional critical factors influencing WBE performance include frequent (e.g., daily) sampling, rapid sample transport, streamlined analytical workflows, and timely result reporting.

Additionally, we compared the mean SARS-CoV-2 RNA concentrations from each dormitory for the Spring semesters of 2021–2023. The analysis performed using the Kruskal–Wallis test revealed no statistically significant differences (F = 0.86, *p* = 0.569) (Appendix A Table A2). This indicates that the contributions of RNA concentrations observed in each dormitory were similar, as also illustrated in Figure 3b.

A Spearman’s correlation analysis was carried out between the positive dormitory infection case count and SARS-CoV-2 RNA concentrations corresponding to the dormitory in 2021, as shown in Figure 4a,b, and separately for each dormitory, as shown in Appendix A Figure A2. Our dataset indicates a small but significant correlation between infection rates reported per dormitory and RNA copies in 2021 (Rs^2^ = 0.03 *p* = 0.006).

Moreover, the mixed linear model applied to the wastewater system for predicting the number of clinical cases reported per building on campus in 2021 was unsuccessful in providing an epidemiological interpretation (*p* = 0.704). In Spring 2022–2023, a Spearman’s correlation between clinical cases (reported by each month) and RNA concentrations was carried out. A weak and non-significant relationship was observed (R_s_^2^ = 0.007, *p* = 0.65). This outcome is not surprising given that numerous off-campus students did not contribute to the campus wastewater system corresponding to particular dormitories under study, yet some of them were still subjected to clinical testing in accordance with mandated testing policies and protocols. Moreover, the clinical health center at NCA&T did not report the cases per dormitory as it had done previously in 2021. The relationship between viral concentration in wastewater and the number of positive clinical cases appears to be strongly influenced by population size and the temporal resolution of the data, suggesting that broader temporal scales may obscure finer patterns [24,48]. Other wastewater surveillance studies have observed that sampling wastewater from neighborhood sites within a campus or individual buildings, compared to cluster of buildings [19], county, or state wastewater treatment plants [12,25], may offer more accurate representations of the viral signal. Therefore, to improve monitoring and avoid potential misinterpretation due to transitions from campus to city, tracking individuals who test positive through wastewater analysis could help maintain accurate city-wide epidemiological trends. Alternatively, clinical testing may lag wastewater monitoring during periods of high incidence (overwhelming contact tracing and clinical testing capacities, or during pandemic peaks) [33,49]. Moreover, as the distribution of home test kits evolved in subsequent years, it became increasingly challenging to monitor more effectively on a per-dormitory basis. To improve clarity, administering questionnaires in these dormitories could provide a more accurate assessment of infection rates. This approach would help establish a comprehensive database to track re-emerging infections, evaluate vaccination coverage, and identify unreported cases from at-home test kits. Ultimately, this data could enhance contact tracing efforts through digital technology [50]. Correlating these data with the daily RNA copy numbers detected could yield more robust and informative results in future studies. Thus, by computing and plotting the total concentration detected daily, it is possible to examine the spread of the virus in a given area for every new infection arising using each case reported in the corresponding dormitory. Another useful parameter would be to conduct a geospatial analysis to evaluate the influence of dormitory age on the transmission dynamics of the virus. This investigation could reveal whether recently constructed dormitories, which are likely to have advanced ventilation systems, exhibit differences in infection rates compared to older facilities. Comprehensive analysis and ongoing monitoring are essential to better understand the seasonal dynamics of viral prevalence in wastewater.

### 3.3. Normalizing SARS-CoV-2 RNA Concentrations by PMMoV Does Not Substantially Impact Results

For Spring 2022–2023, we generated PMMoV data at the NCA&T campus to determine if SARS-CoV-2 RNA concentrations normalized using these data improved correlations to clinical data. Our study for the period of Spring 2022–2023 showed that normalizing SARS-CoV-2 concentrations by PMMoV yielded low, insignificant correlation (Rs^2^ = 0.002, *p* = 0.29). In addition, trends for normalized RNA concentrations with clinical cases in Spring 2022–2023 had a weak and insignificant correlation (Rs^2^ = 0.02, *p* = 0.85). This suggests that PMMoV normalization did not significantly enhance the correlation with clinical data compared to non-normalized SARS-CoV-2 concentrations (Figure 5b).

The use of surrogate viruses as controls for fecal strength has been demonstrated to ensure consistent method performance across sample types and over time, addressing changes in target concentration efficiencies and RNA extraction [12,25,51]. While some studies advocate for PMMoV normalization [52,53], others have found no improvement in using a normalization fecal marker to positively affect correlations between wastewater concentrations and positive clinical cases [39,54,55,56,57]. Zhan et al. [40] demonstrated that β2M (β-2 microglobulin), which copurifies with SARS-CoV-2 RNA and is present in various body fluids, provided more promising results compared to PMMoV [58,59,60]. Additionally, Hsu et al. [61] identified the CAF metabolite PARA as a more reliable biomarker than PMMoV due to its higher accuracy and temporal consistency in reflecting population dynamics and dilution in wastewater. Maal Bared et al. [56] also showed that using unnormalized concentrations or normalizing by other parameters such as total Kjeldahl nitrogen (TKN), total phosphorus (TP), and ammonia (NH3) provided comparable results to PMMoV. Consequently, the use of alternative fecal biomarkers should be considered in enhancing the accuracy of SARS-CoV-2 quantification in wastewater and improve estimates of human case counts.

### 3.4. Comparison of Composite and Grab Sampling Methods on NCA&T Campus

To identify the optimal sampling method correlating with community transmission of SARS-CoV-2, samples were collected using two methods: Grab and Composite sampling since Fall 2021. The comparison of Grab and Composite samples was conducted for two dormitories, Aggie Village 4 and Barbee Hall. A strong positive and significant correlation was observed between Grab and Composite sampling methods for both halls: Aggie Village 4 (Grab) and Aggie Village 4-Auto (Composite) samples (Rs^2^ = 0.84, *p* = 0.00) and between Barbee (Grab) and Barbee-Auto (Composite) samples (Rs^2^ = 0.80, *p* = 0.00) between Fall 2021 and Spring 2023, as shown in Figure 6a. Our findings align with previous studies that have reported a strong significant correlation between both methods [62,63]. A study conducted in Virginia found a strong correlation between Grab and Composite samples during a period of low transmission, based on samples collected from a municipal wastewater treatment facility [64]. In contrast, a study by Rafiee et al. [65] reported that grab samples exhibited lower levels of SARS-CoV-2 RNA in wastewater compared to Composite samples. This disparity in performance is likely attributable to the substantial daily fluctuations in human fecal matter at the sub-sewer shed level. Moreover, Composite sampling, which aggregates multiple aliquots over a week, may capture a more representative average of viral presence compared to Grab sampling, which can introduce temporal variability. Increased frequency of Composite sampling might enhance accuracy in SARS-CoV-2 concentration estimates. To mitigate variability, it is suggested that Grab samples be collected multiple times per day, potentially up to six times, to improve correlation between viral shedding and wastewater data.

Furthermore, a correlation analysis was conducted to compare the two sampling methods using normalized data adjusted by PMMoV as shown in Figure 6b. For the Aggie V4 (Grab) and Aggie V4-AS (Composite) methods, a strong and significant correlation was observed (Rs^2^ = 0.79, *p* = 0.00). Similarly, for the Barbee (Grab) and Barbee-Auto (Composite) hall, a strong and significant correlation was observed (Rs^2^ = 0.78, *p* = 0.00). The incorporation of PMMoV did not result in a significant enhancement in the correlations between the two sampling methods. Variations in seasonal patterns of viral biomarkers attributable to differences in sewer shed characteristics (e.g., flow rates, input sources) may influence the efficacy of normalization techniques in enhancing correlations [66]. Normalization of SARS-CoV-2 concentrations using PMMoV as a reference may provide benefits at certain sites or times by accounting for variations in flow, fecal matter, or sampling methodologies. However, caution is warranted in interpreting normalized data. The behavior of PMMoV in the sewer system may differ from that of SARS-CoV-2, with the latter being more uniformly distributed between solids and supernatant [66,67]. Consequently, increases in wastewater volume due to environmental factors such as precipitation or snowmelt are also likely to result in a dilution of viral concentrations of the biomarker [68].

## 4. Conclusions

This study represented the first and one of the most comprehensive wastewater surveillance efforts on the campus of North Carolina A&T State University across 11 dormitories over a period of 3 years. In this study, we have demonstrated that WBE is a novel method as it provides a standardized detection of viruses and pathogen load in wastewater sampling on campus. This technology is innovative, helping to detect and monitor the spread of SARS-CoV-2 on the campus of North Carolina A&T State University between Spring 2021 and Spring 2023 while allowing the school administration to engage in critical conversations regarding topics such as dormitory occupancy, clinical testing, and switching to in-person classes versus online mode. Through this study, informed decisions about the health of NCAT students and staff while also continually monitoring the risk of COVID-19 spread was accomplished (particularly at the peak of the pandemic in 2021).

Wastewater-based epidemiology (WBE) has significant potential as an effective early warning system for community-wide monitoring of viruses in public health surveillance, as shown in other studies [5,12,19,26,69]. While retrospective studies have highlighted its promise for early detection and real-time early warning. In this study, sampling at the building or dormitory level introduced higher stochastic variability compared to retrospective studies that primarily focused on wastewater treatment plants (WWTPs) and local sewer sheds, which demonstrated stronger correlations with clinical case data. While our study employed WBE for SARS-CoV-2 surveillance, its potential extends beyond the scope of this paper and can be applied to monitoring other respiratory and enteric viruses [70,71,72].

Our study identified several challenges that present opportunities for further research, particularly in campus settings. A key limitation was the confounding influence of clinical data variations. While a small but significant correlation (R^2^ = 0.03, *p* = 0.006) was observed between SARS-CoV-2 RNA concentrations and weekly reported cases in 2021, this relationship weakened in Spring 2022–2023 (R^2^ = 0.007, *p* = 0.65). A mixed linear model also failed to predict case numbers (*p* = 0.704), highlighting the need for expanded dormitory sampling and improved case reporting. Normalization using PMMoV to account for fecal dilution did not significantly improve correlations between RNA concentrations and clinical cases (*p* > 0.005), consistent with other studies suggesting variations in recovery efficiency and wastewater chemistry. However, Grab and Composite sampling methods showed a strong correlation at Aggie Village 4 (R^2^ = 0.84, *p* = 0.00) and Barbee Hall (R^2^ = 0.80, *p* = 0.00), supporting their robustness.

To enhance WBE’s reliability for public health surveillance, maintaining consistent analytical performance is essential despite its inherent variability in wastewater sampling. Future studies should explore infectious virion quantification in wastewater to better assess viral decay and transmission trends in sewers. Therefore, with robust quality assurance and quality control protocols, WBE can serve as a valuable complement to, but not a replacement for, clinical testing.

## Figures and Tables

**Figure 1 microorganisms-13-00924-f001:**
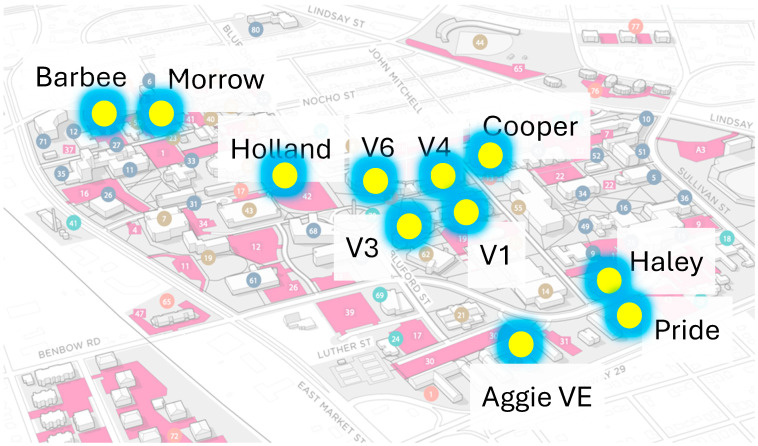
Campus map of North Carolina A&T State University with marked dormitories for sample collection (retrieved and modified from Campus website).

**Figure 2 microorganisms-13-00924-f002:**
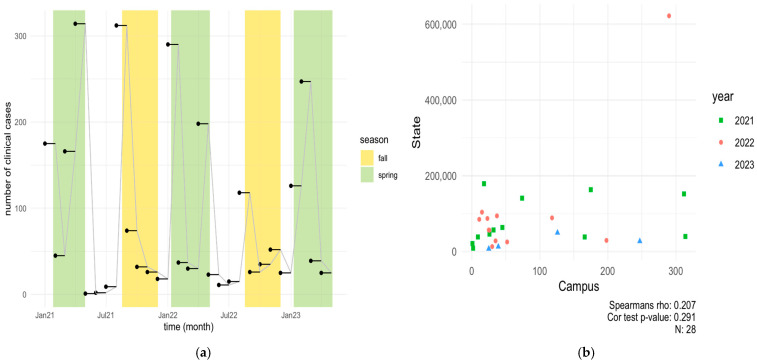
(**a**) Clinical cases on NCA&T campus between Spring 2021 and Spring 2023; green bars represent Spring semesters and yellow bars represent Fall semesters. (**b**) Scatter plot showing the correlation (Rs^2^ = 0.04, *p* = 0.29) between the total number of clinical cases in NC state and NCA&T campus between Spring 2021 and Spring 2023.

**Figure 3 microorganisms-13-00924-f003:**
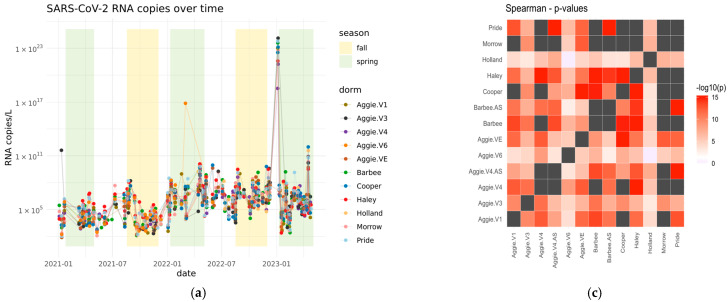
(**a**) SARS-CoV-2 RNA copies between Spring 2021 and Spring2023. The yellow area represents the Spring semesters, and the green area represents the Fall semesters. (**b**) Histogram of Spearman’s rho for all pairwise correlations between dormitories. (**c**) Heatmap of *p*-value on negative log 10 scale of the correlation between each pair of dormitories as assessed using the spearman’s correlation test. (**d**) Heatmap representation of the pairwise spearman’s rho corresponding to the *p*-values in (**c**) and the distribution in (**b**). The high correlation values indicate similar contributions of RNA concentrations among the dormitories, with no significant differences observed. This suggests a consistent distribution of SARS-CoV-2 RNA levels across sampling sites over the analyzed period.

**Figure 4 microorganisms-13-00924-f004:**
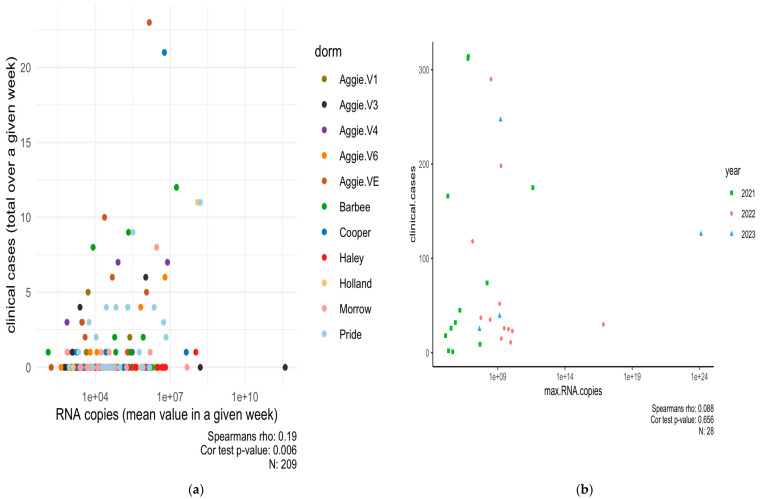
(**a**) Predictions of the incidence rates based on observed amount of N1 gene copy numbers in WW in selected buildings for 2021 (Rs^2^ = 0.03, *p*-value = 0.006). (**b**) Scatter plot between NCA&T campus-wide clinical cases with mean of SARS-CoV-2 RNA concentrations in Spring 2022–2023 on a log scale (Rs^2^ = 0.007, *p*-value = 0.65).

**Figure 5 microorganisms-13-00924-f005:**
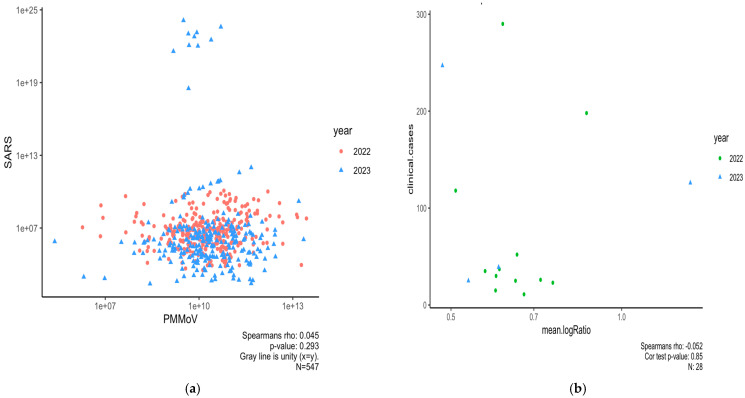
(**a**) Scatter plot between PMMoV and SARS-CoV-2 RNA concentrations for Spring 2022–2023 (Rs^2^ = 0.002, *p*-value = 0.29). (**b**) Scatter plot showing correlation between clinical cases and normalized RNA concentrations by PMMoV in 2022–2023; mean.logRatio represents mean of “log10(SARS RNA copies)/log10(PMMoV RNA copies).

**Figure 6 microorganisms-13-00924-f006:**
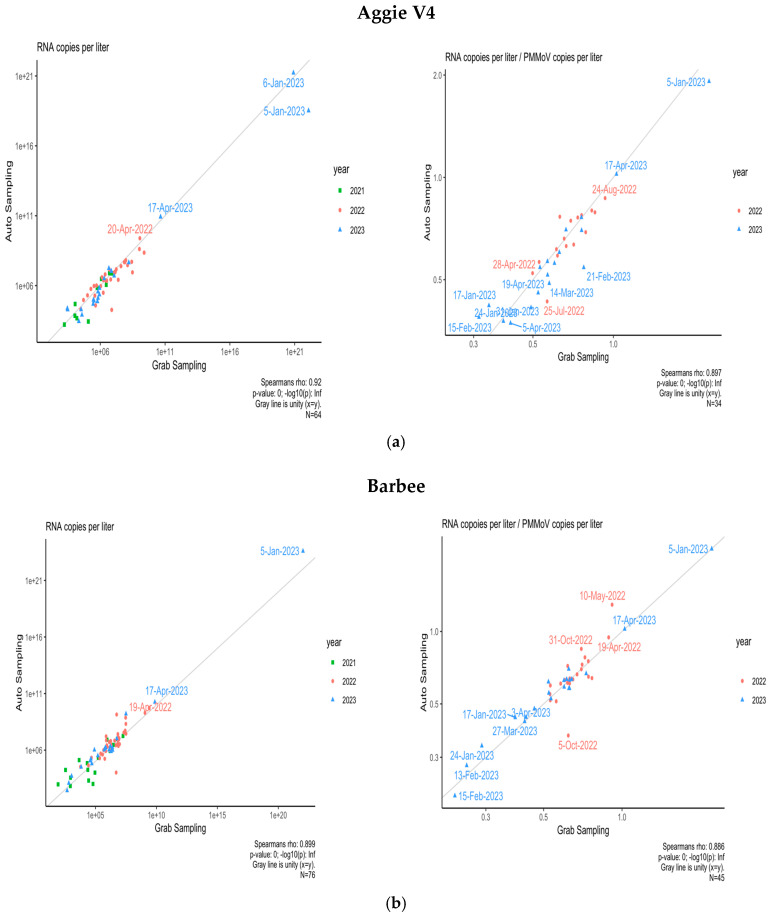
(**a**) Spearman’s correlation analysis between SARS-CoV-2 RNA concentrations of two sampling methods (Grab and Composite) for Aggie V4 hall and normalized by PMMoV. (**b**) Spearman’s correlation analysis between SARS-CoV-2 RNA concentrations of two sampling methods (Grab and Composite) for Barbee Hall and normalized by PMMoV.

## Data Availability

The original data presented in the study are openly available in the project git repository at https://github.com/FodorLab/SARS-CoV-2_wastewater_NCAT (with the release version of publication made on 8 April 2025 corresponding to this publication) and the Code Ocean capsule doi:10.24433/CO.8844928.v1, or by request from the corresponding author.

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
