# Peer review of "Wastewater Speaks: Evaluating SARS-CoV-2 Surveillance, Sampling Methods, and Seasonal Infection Trends on a University Campus"

_microorganisms, 2025, doi:10.3390/microorganisms13040924_

Round 1
Reviewer 1 Report
Comments and Suggestions for Authors
In the abstract, in the keywords section, I suggest changing the keywords that are in the article title. The same words used in the article title are not usually used in the keywords;
In the methodology section, section 2.3, describe how the 200mL of wastewater were collected, type of bottle, type of transport, temperature of the water collected, temperature at which the water was transported, etc.
In the methodology section, section 2.5. Real Time-quantitative Polymerase Chain Reaction, state the concentration of the genetic material that was used to perform the qPCR.
In the methodology section, which negative control was used in the qPCR reaction?
I suggest adding references from the last 3 years, 2023, 2024 and 2025.
In the results section, I suggest increasing the font size of figures 3a, 3b and 3d.
Author Response
Comment 1: In the abstract, in the keywords section, I suggest changing the keywords that are in the article title. The same words used in the article title are not usually used in the keywords-
Response 1: Thank you for your valuable feedback. We agree with your suggestion and have adjusted the keywords by removing those that are already included in the article title.
Comment 2: In the methodology section, section 2.3, describe how the 200mL of wastewater were collected, type of bottle, type of transport, temperature of the water collected, temperature at which the water was transported, etc.-
Response 2: Thank you for this comment. We have revised the methods to include type of bottle, type of transportation, temperature of water during transportation and storage, and is highlighted on page 4, line numbers 141-146.
Comment 3: In the methodology section, section 2.5. Real Time-quantitative Polymerase Chain Reaction, state the concentration of the genetic material that was used to perform the qPCR. -
Response 3: Thank you for this comment. The concentration of the genetic material used for qPCR was not directly measured. However, a SARS-CoV-2 and PMMoV (internal control) standard curve was used to ensure accurate quantification and validation of the assay.
Comment 4: In the methodology section, which negative control was used in the qPCR reaction?
Response 4: Thank you for your comment. Non-template control (NTC) and nuclease-free water were included as negative controls to monitor for contamination and ensure assay reliability. We have mentioned this in methods section 2.5 on page 4.
Comment 5: I suggest adding references from the last 3 years, 2023, 2024 and 2025.
Response 5: Thank you for your suggestion. Our manuscript includes the most recent references available at the time of submission. We have also added more recent 2024 references in the conclusion section at lines 496-498 on page 14.
Comment 6: In the results section, I suggest increasing the font size of figures 3a, 3b and 3d.
Response 6: Thank you for your suggestion. We have enlarged the figure dimensions for improved visualization. The font is Times New Roman, size 10.
Reviewer 2 Report
Comments and Suggestions for Authors
This study monitored the prevalence of SARS-CoV-2 on the campus of North Carolina State University from 2021 to 2023 through wastewater epidemiology (WBE). The study collected wastewater samples from 11 dormitory building drainage outlets, used RT qPCR to detect viral RNA concentration, and compared them with clinical case data. The results showed a strong correlation between Grab sampling and Composite sampling methods (R ²=0.84 and 0.80). The viral RNA copy number in the spring semester was higher than that in the autumn and summer semesters, and the viral concentration significantly increased after students returned to school. Research still indicates that WBE has significant value in campus infectious disease surveillance, but further optimization is needed to enhance its reliability as a supplementary tool for clinical testing. This study has certain practical value.
Here are my comments and suggestions:
1,In the MATERIALS AND METHODS section, specific criteria for selecting 11 designated dormitories should be supplemented. In addition to population size, factors such as dormitory location distribution and building type can be considered for their impact on wastewater monitoring.
2,In the RESULTS section, for the analysis of RNA concentration differences between different dormitories, in addition to conducting statistical tests, specific data characteristics and trends of dormitories with significant differences can be visually presented.
3,The study did not quantify infectious virus particles in sewage samples. If relevant experiments cannot be supplemented, it is recommended to discuss them as a limitation in the discussion. Because it can provide more information about the dynamics of virus transmission and the natural decay of viruses in sewers.
4,The study only focused on monitoring SARS-CoV-2 in wastewater. Suggest exploring the potential of wastewater epidemiology in monitoring other viruses, such as enteroviruses, influenza viruses, or norovirus.
Author Response
Comment 1: In the MATERIALS AND METHODS section, specific criteria for selecting 11 designated dormitories should be supplemented. In addition to population size, factors such as dormitory location distribution and building type can be considered for their impact on wastewater monitoring.
Response 1: Thank you for your comment. The selection of dormitories was based on their large population size and the strategic placement of manholes just outside, leading to a significantly higher wastewater inflow rate, as outlined in Methods Section 2.1. To enhance clarity, we have included Supplementary Table A.1, which details the sampling sites, frequency, and sampling type. The dormitories are classified as residential buildings within the campus. While we do not have precise population figures for each dormitory, we selected 11 out of the 16 on campus, excluding four due to insufficient wastewater inflow for collection. Additionally, the selected dormitories remained open to students during the COVID period and had dedicated manholes outside the buildings, enabling specific and accessible sampling.
Comment 2: In the RESULTS section, for the analysis of RNA concentration differences between different dormitories, in addition to conducting statistical tests, specific data characteristics and trends of dormitories with significant differences can be visually presented.
Response 2: Thank you for your comment. SARS CoV-2 RNA trends for each dormitory and incidence rate with RNA concentrations for each dormitory are shown in Supplementary figures A.1, and A.2.
Comment 3: The study did not quantify infectious virus particles in sewage samples. If relevant experiments cannot be supplemented, it is recommended to discuss them as a limitation in the discussion. Because it can provide more information about the dynamics of virus transmission and the natural decay of viruses in sewers. -
Response 3: Thank you for your suggestion. We did not measure infectious virus particles in wastewater samples for several reasons including the requirement of a BSL 3 lab which we do not have access to. We have discussed the importance of this data and suggested as a possibility for future studies, in our conclusion section on page 14, highlighted in yellow.
Comment 4: The study only focused on monitoring SARS-CoV-2 in wastewater. Suggest exploring the potential of wastewater epidemiology in monitoring other viruses, such as enteroviruses, influenza viruses, or norovirus.
Response 4: Thank you for pointing this out, we have discussed this information in our literature review (Introduction section, page 2, lines 58-59, highlighted in yellow) and conclusion section (page 14, highlighted in yellow).
Reviewer 3 Report
Comments and Suggestions for Authors
The manuscript "Wastewater Speaks: Evaluating SARS-CoV-2 Surveillance, Sampling Methods, and Seasonal Infection Trends on a University Campus" shows interesting results over a three-year period comparing the SARS-CoV2 RNA concentrations in wastewater with clinical cases in a University campus. The manuscript is overall well-structured and written.
Specific comments:
1.Abbreviations like STIs should be defined when first mentioned.
2.The number of samples analyzed is not clear, I suggest authors to include a Table summarizing the simple site, frequency, number of samples and type of sample (grab or composite) collected in each case for better clarity.
3.The composite sample were from 24-hr? was an autosampler employed or was it manually collected? I suggest authors to include this information in the manuscript.
4.The RNA standard curve for SARS-CoV2 is included in the detection kit? Please specify.
5.Figure 3 caption lacks a more detailed description.
6.The conclusion´s section is excessively long, most of it could be included in the results and discussion section. The conclusions should be more concise focusing only on the relevant points of the work.
Author Response
Comment 1: Abbreviations like STIs should be defined when first mentioned.
Response 1: Thank you for this comment. The required change has been made and highlighted on page 2, line 56.
Comment 2: The number of samples analysed is not clear, I suggest authors to include a Table summarizing the simple site, frequency, number of samples and type of sample (grab or composite) collected in each case for better clarity
Response 2: Thank you for this comment. We have included a table that summarizes the sample site, sample type, sample frequency and number of samples in Supplementary
Comment 3: The composite sample were from 24-hr? was an autosampler employed or was it manually collected? I suggest authors to include this information in the manuscript.
Response 3: Thank you for this question. Yes, the composite sample reflected a 24-hour sampling period. We have included a description of the autosamplers used and the frequency of sample collection in the Methods section 2.1, lines 117-121 highlighted in yellow on page 3.
Comment 4: The RNA standard curve for SARS-CoV2 is included in the detection kit? Please specify.
Response 4: Thank you for your question. Yes, the SARS-CoV-2 RNA standard curve was included in the SARS-CoV-2 RT-qPCR Kit for Wastewater (Promega; Madison, MI, USA) and is described in the Methods section 2.6, lines 180-185 on page 4.
Comment 5: Figure 3 caption lacks a more detailed description.
Response 5: Thank you for your comment. We have added a more detailed description to figure 3 legend. We have highlighted it in yellow on page 8, line 301-307.
Comment 6: The conclusion´s section is excessively long, most of it could be included in the results and discussion section. The conclusions should be more concise focusing only on the relevant points of the work.
Response 6: Thank you for your suggestion. We agree with your comment. We have significantly condensed the conclusion section and incorporated some of the previous content into the discussion section.
Reviewer 4 Report
Comments and Suggestions for Authors
The introduction is a very good introduction to the discussed issue.
The methodology is presented in great detail, it contains a description of all the procedures used.
In Figure No. 1, please move the markings of the sampling locations - No. 4 is not very visible.
The results are presented very well. Well discussed, divided thematically into subchapters.
The summary chapter is, in my opinion, too extensive. It even contains elements of discussion and citations of literature. I would suggest moving fragments of the text to the results and discussion chapter. And in this part, leave only the summary - the most important conclusions from the conducted research.
Literature relevant to the subject. Most of the literature is from after 2020. And this is understandable, because since this year we have felt the problem of the SARS-CoV-2 virus all over the world.
Author Response
Comment 1: The introduction is a very good introduction to the discussed issue.
Response1: Thank you for your comment.
Comment 2: The methodology is presented in great detail; it contains a description of all the procedures used.
Response 2: Thank you for your comment.
Comment 3: In Figure No. 1, please move the markings of the sampling locations - No. 4 is not very visible.
Response 3: Thank you for your suggestion. We have revised the sample locations on the map to improve visibility in Figure 1.
Comment 4: The results are presented very well. Well, discussed, divided thematically into subchapters.
Response 4: Thank you for your comment
Comment 5: The summary chapter is, in my opinion, too extensive. It even contains elements of discussion and citations of literature. I would suggest moving fragments of the text to the results and discussion chapter. And in this part, leave only the summary - the most important conclusions from the conducted research.
Response 5: Thank you for your suggestion. We agree with your comment. We have significantly condensed the conclusion section and incorporated some of the previous content into the discussion section.
Comment 6: Literature relevant to the subject. Most of the literature is from after 2020. And this is understandable, because since this year we have felt the problem of the SARS-CoV-2 virus all over the world.
Response 6: Thank you for your comment.
Round 2
Reviewer 2 Report
Comments and Suggestions for Authors
The author has made detailed revisions, and I have no other comments.
Reviewer 3 Report
Comments and Suggestions for Authors
All comments were addressed.